# Energy Internet Opportunities in Distributed Peer-to-Peer Energy Trading Reveal by Blockchain for Future Smart Grid 2.0

**DOI:** 10.3390/s22218397

**Published:** 2022-11-01

**Authors:** Bassam Zafar, Sami Ben Slama

**Affiliations:** 1Information System Department, FCIT King Abdulaziz University, Jeddah 22254, Saudi Arabia; 2The Applied College, King Abdulaziz University, Jeddah 22254, Saudi Arabia; 3Analysis and Processing of Electrical and Energy Systems Unit, Faculty of Sciences of Tunis El Manar, Tunis 2092, Tunisia

**Keywords:** SG 2.0, energy internet, prosumer, peer-to-peer, blockchain, energy trade

## Abstract

The Energy Internet (EI) and Smart Grid 2.0 (SG 2.0) concepts are potential challenges in industry and research. The purpose of SG 2.0 and EI is to automate innovative power grid operations. To move from Distribution Network Operators (DSO) to consumer-centric distributed power grid management, the blockchain and smart contracts are applicable. Blockchain technology and integrated SGs will present challenges, limiting the deployment of Distributed Energy Resources (DERs). This review looks at the decentralization of the Smart Grid 2.0 using blockchain technology. Energy trading has increased due to access to distributed energy sources and electricity producers who can financially export surplus fuels. The energy trading system successfully combines energy from multiple sources to ensure consistent and optimal use of available resources and better facilities for energy users. Peer-to-peer (P2P) energy trading is a common field of study that presents some administrative and technical difficulties. This article provides a general overview of P2P energy exchange. It discusses how blockchain can improve transparency and overall performance, including the degree of decentralization, scalability, and device reliability. The research is extended to examine unresolved issues and potential directions for P2P blockchain-based energy sharing in the future. In fact, this paper also demonstrates the importance of blockchain in future smart grid activities and its blockchain-based applications. The study also briefly examines the issues associated with blockchain integration, ensuring the decentralized, secure and scalable operation of autonomous electric grids in the future.

## 1. Introduction

Nowadays, SGs 2.0 are regarded as attractive study topics due to their effectiveness in overcoming previous networks’ calamitous concerns and ambiguities. Incorporating local networks and dispersed energy supplies, SG technologies considerably reduce energy demand. Existing infrastructure operators are motivated by the rapid increase in electricity consumption and public infrastructures [1]. SGs have also been selected as effective self-processing technologies that permit the transfer of electricity and innovative technologies in the same ways. Diverse kinds of energy consumption that employ resources, management, distribution, and interchange have been developing rapidly and efficiently [2]. This new approach to energy use is known as “Prosumer”. In [3], the authors stated that Prosumers have a considerable interest in SG as a result of the flexibility and energy efficiency of the electricity distribution phase. Indeed, the Prosumer will play a critical role in emerging SGs by coordinating peak periods, energy demand and optimization [4].

Consequently, it is anticipated that Energy Management Systems (EMSs) and Internet of Things (IoT) systems will be incorporated into the identification and analysis of relevant issues, as well as the implementation and testing of the impact of the prosumer’s requirements on possible SGs. These factors have motivated the Power Service Provider (PSP) to enhance power lines by providing advanced technology and applications for consumer interactions, responding to Prosumer needs and implementing the energy leasing technique [5]. This methodology motivated PSP to incorporate scalability, modern software applications, decentralized structures and analytical measures. The latter manages services appropriately, engages effectively and achieves business goals [6]. Therefore, as opposed to leasing power lines, the ISP provided and secured power lines for its clients [7]. In [8], the authors proved that SG was also certified and selected as a solution that encompasses fuels, knowledge, communications, commercial domains and various applications to achieve scientific, economic and legal aims. Small- to Medium-Sized Businesses rely more on convergence SG technologies, interoperability and new specifications due to digitalization (SMEs). Innovative technology and sophisticated applications have transformed the legacy network. As a result, the global energy infrastructure has become more diverse, resulting in digital exchanges between all stakeholders, including industries and Prosumers [9]. In [10], the author highlights how most innovations should be fully interoperable with PSP and businesses, with their use as a potential solution. Emerging potential solutions included many technologies, such as smart homes, artificial manufacturing, smart cities and improved industry 4.0 applications [11]. Therefore, the specific qualifications and characteristics of high-end technologies, consumer applications and intelligent applications are essential, as they will determine future classifications of applications, tools and technologies used in smart homes, Internet infrastructure, web services, computers and infrastructure [12].

In [13], the authors highlight that the IoT will revolutionize the conception of our global communities. According to the International Energy Agency (*IEA*), it is developing innovative grid technology that enhances and replaces existing systems and propels individuals and communities towards innovative infrastructure developments. Public security is one of the pillars of constructing a more stable, productive and secure economy. The latter will enable the management and resolution of all lighting, traffic light, pollution, parking, street alarm, early resource detection, emergency weather and energy storage issues. SGs provides the same for infrastructure, power lines, smart meters, components, post-station systems, switches, sensors and other applications [14]. SG has become less expensive than the current electrical grid due to the diversity of innovative technology. Switching to SG necessitates electricity from numerous, widely scattered sources [15]. The advanced grid development will incorporate traditional power plants, solar and wind energy sources, plug-in devices and energy storage facilities. Using and keeping data can drastically cut energy use and expenditures. For instance, in [16], the authors illustrate that intelligent lighting is designed to track across different regions automatically, is adapted to accommodate daylight or traffic requirements and can rapidly assess energy demand. In [17], the authors show that consumers can alter house temperatures and air conditioners while at work or on holidays. According to the authors, SGs reduce expenses through tracking, intelligent energy and switching sources when power failures are detected. In [18], the authors argue that expansion of the IoT will encourage the US energy sector to incorporate renewable facilities to increase wind power generation, micro-grid networks and feeding structures. In [19], the authors showed that EI and SG 2.0 might enable the transportation and parking station sectors to connect and collect real-time data from drivers and authorities. This vision will minimize road congestion, enhance traffic solutions, report pedestrian street collisions, damage the urban environment and automatically encourage road charges and parking meters. Internet of Things technology lets autonomous vehicles operate wirelessly. In [20], the authors reported that IoT technologies can control waste and water and cut greenhouse gas emissions. This includes real-time product monitoring and loss management results. In [21], the authors illustrate how IoT and big data monitor water movement and temperature, manage energy demand and reduce waste. Timers and infrastructure help achieve these goals. In [22], the authors state that the IoT would be utilized to distribute electricity to low-population areas by linking national or regional infrastructure. These networks are needed to use modern energy technology. In [23], the authors suggest the IoT as the most excellent answer for intelligent cities and SGs. This letter will indicate real-time regional difficulties. In Mannheim, Germany, it was used to implement SG applications. Mannheim’s cities implemented green resources with this project and planned and developed energy usage [24]. Schneider Electric offers wired solar power systems in Mannheim. This will allow families to have access to PV systems and control and maintenance equipment until the entire network is depleted or solar energy is produced and converted to satisfy peak demand [25].

In [26], the authors report that the Lumin Energy Project (LEP) used the IoT to reduce expenditures and emissions and promote renewable energy. This project offers an efficient storage program for solar panels. In [27], the authors note that various techniques have recently been developed to decentralize messaging, data storage and transportation. This phenomenon encourages academics and the industry to consider edge computing as a solution. Computing networks will feature edge computing. This will connect cloud storage, networks and databases. In [28,29], the authors suggested that edge computing would enhance reaction times, reduce energy rates and increase interactions, scalability and confidentiality. Edge computing helps heterogeneous IoT systems communicate through unique network topology and different devices (sensors, cars, machines, computers, gauges, etc.). In [30], the authors state that business 4.0, energy management and consumers will benefit from IoT SG technologies. Heterogeneous IoT data can be leveraged to produce infrastructure and advanced computing technology solutions (IoT sensors, cloud services, edge nodes) [31]. IoT platforms may obtain information from hundreds or thousands of data sets using Edge Computing, which helps organizations decide whether it will work and predict how emerging technology can affect culture. Many analysts will plan the power grid in light of hybrid electric vehicles [32]. In [33], the authors report that SG responded to common concerns about electricity consumption by integrating wireless detection sensors and cloud computing. The use and distribution of electrical data in research groups raise data protection considerations. The blockchain platform strengthens the foundations of SG. The Blockchain infrastructure will have proper ways to oversee the exchange of customer data with SG [34].

The contributions of this work to others are summarized in Figure 1. These works focus on one topic or some of them. There is no comprehensive study on blockchain testing in the SG industry. We decided to publish an entire post on blockchain research in smart grids. The main contributions of this paper, in contrast to related survey work, are as follows:We define blockchain technology, smart contracts, blockchain classes and blockchain consensus procedures to deliver SG 2.0.We provide a list of prerequisites for SG 2.0. We show how P2P energy, privacy and trust commerce can be handled via blockchain.To explain why blockchain technology can be used and how it can deal with these issues, we discuss the main research challenges of the various components and scenarios of the smart grid.We are looking into the possibilities of blockchain in SG 2.0.We evaluate existing blockchain technologies based on SG 2.0. We also draw attention to the issues discussed and the applications of the blockchain.We discuss SG 2.0 and the idea of the blockchain.The problems and future directions of research are determined through our study.We outline open issues, challenges and future research directions related to blockchain smart grids.

A list of acronyms used throughout the paper is presented in Abbreviations. The remainder of the survey is organized as follows: Section 1 introduces the SG 2.0 and IoT concepts. In Section 2, “Preliminaries: Detailed Analysis of the Literature”, we will provide an overview of the state-of-the-art of essential studies that address various challenges and issues in SGs. Section 3, “Energy Domain Prosumer Classifications”, discusses the peer-to-peer energy trading concept and its architecture and techniques. In Section 4, “Blockchain Technology in SG 2.0”, we discuss the blockchain architecture, the information processing in Prosumer SGs and the concept and models of future energy management systems. In Section 5, we provide open issues and future directions. We conclude the survey paper in Section 6.

## 2. Preliminaries: Detailed Analysis of the Literature

### 2.1. SG Domains

SGs are tools used to install network infrastructure in households, companies and the network, as well as to control power consumption and other factors. SG innovations are self-contained buildings that can address power grid issues and guarantee consistent electricity for all customers. With cleaner and more energy-efficient, reliable and sustainable electricity, SGs will replace the current networks [35]. This section illustrates the general hierarchical distribution structure of blockchain in SG 2.0, as seen in Figure 2. The future Smart Grid 2.0 hierarchy and pills are:*a.* *Renewable Energy;**b.* *Electrical Vehicle (EV);**c.* *P2P Energy Trading;**d.* *Energy Data Managemnt;**e.* *Microgrids;**f.* *Grid Distribution Network.*

SG service providers would presume that they are contributing to the development of the public network infrastructure due to the network’s complexity [36]. The modern infrastructure in Singapore reduces electricity use. Performance can be enhanced by upgrading the sensing rates and continuous power [37]. In [38], the authors recommend altering current methodology and practices to stop production, energy demand and consumer problems. The active service should be improved by a few standards and problems. In [39], the authors stated that selecting the supplier unit with the lowest price that has system controls improves operational production. By altering their energy consumption and purchases, customers can use SG to affect demand trends. This pattern will increase consumer interest and energy sales. SG was chosen as an appealing new technology because it offers up-to-date details on energy use, service delivery, and advantages. According to the authors’ description in [40], SG augments customers’ access to local energy services by combining wind, heat, power and carbon efficiency. In [41], the authors demonstrate that SG produces energy at various scales (and costs). Customers can select from competing bidders in marketplaces that are well-designed and administered. Markets may manage these variables efficiently. According to operational and economic realities, regulators, creditors and customers can amend corporation law. Electricity, power, location, times, levels and performance are real-time network information. Modern SGs viewed electricity purchasing as a pillar and objective. In [42], the authors proved the electricity purchasing consideration. This proof relies on an innovative infrastructure that integrates policy and customer service. It monitors light, faults, artificial sources and energy demand. Future smart grids challenge energy demand and dispersed production. SG end-users can play both consumer and producer (so-called “Prosumers”) (See Table 1). Future network services may include prosumers to help SGs overcome difficulties and challenges [43]. Prosumer Groups (ProG) aim to transform traditional customers into productive consumers, increase SG performance, and provide an economical, logistical and sustainability advantage [44]. In [45], the authors mentionned that prosumers want to generate and consume energy and share additional power with other customers in the distribution system via edge technology. In the early stages of smart grid deployment, it is important to understand consumers’ responsibilities and priorities to maximize the use of edge technologies, procedures, business models and growth incentives [46]. In [47], the authors chose Prosumer SG functions and attributes depending on energy consumption tactics. In [48], the authors noted that Prosumer SG functionalities and features are based on pre-created energy consumption techniques. The “engineer” promotes rising technologies; the “Green Prosumer” is interested in novel environmental approaches; and the “value seeker” seeks economic rewards and Prosumer efficiency and consistency. Edge computing is a promising Prosumer technology. This will improve privacy and data protection, operating output, market quality, stability, network management and infrastructure handling, reaction time, data dissemination, device performance and operating costs [49].

The SG is an advanced and asynchronous digital power transmission system that consists of forecasting multiple complications; is self-healing and adaptable; and contributes to sustainable development. SG adoption seems to be growing, energy companies are gaining dynamism through smart meters. Switching to smart grids will change power generation systems that will encourage Prosumers and employees who make energy-related life decisions [50]. Evaluations and research have examined Prosumers’ adoption of the smart grid. There are a few methodological statistics for general interest variables and the number of business and innovation category studies. Previous research works both support and contradict the assumption that customers value rewards. It offers guidelines for many techniques. Due to the diversity of smart grid projects’ power grids, outcomes are considered as specific topic. The SG is a vital aim based on research that has dominated generations of energy users. This investment seeks healthy growth and green energy, among other customer concerns. Current technology’s role on brand awareness was studied [51]. Edge computing IoT architecture should retain motivation for energy management and sustainable use. Prosumer’s SG use will motivate everyone to enhance their strategic goals or obtain independence while enabling the smart grid to minimize costs. Fully automated economic or automated procedures will provide stability and good logistical performance, which is needed. The main problem for implementing smart grid projects is the lack of integration standards for the residential sector, where there are tremendous complications and no similar goals, aims for prosperity, demand habits, requirements, priorities or local restrictions. Edge computing, the IoT and the blockchain must be adopted to govern Prosumers. This survey is based on principles, applications, field surveys and theory. Table 1 outlines each interest area’s level of use. This issue could isolate prosumers. This isolation may cause irregular prices. Even with less local participation and the need for worldwide expansion, the model works.

### 2.2. SG 2.0

#### 2.2.1. The SG 2.0 Concept

SG 2.0 is garnering attention for incorporating Distributed Energy Resources (DERs). The following sections compare SG 2.0 to its predecessor, the first generation of smart grids. SG 2.0 emerged as a renewable energy grid integration facilitator. Using smart meters, SG 1.0 could route and communicate bidirectionally between the Transmission System Operators (TSO) and Distribution Network Operators (DSO) and the users. It worked with the existing electrical infrastructure to turn standard energy meters into telecommunications-compatible intelligent meters (the only change required). With the focus shifting to renewable energy integration to overcome the conventional power generation difficulties, large, small and domestic solar and wind facilities are increasing incentives to encourage sustainable energy use. Consumers now have the choice of the appropriate source for their electricity service provider (PES) [52]. In [53], the authors state that the Energy Internet (EI), also known as ‘SG 2.0’, is a novel concept in future SGs. This is an anticipated electrical system integrating various energy sources and intelligent charges, with supervision and control managed by SG 2.0 protocols over the Internet. It predicts self-controlled, self-optimized and self-healing power grids. SG 2.0 exchanges energy and relevant data smoothly, like the Internet does. In [54], the authors report that SG 2.0 is a next-generation system that uses Internet-based P2P networks to monitor and control diverse energy sources. This enables the integration of renewable energy and energy storage systems into future SGs, plug-and-play electric vehicle charging, real-time monitoring/control of power grids, energy data acquisition/management, and the automation of energy balancing services. This new category of Prosumers/consumers is known as ‘consumers. Distracted consumers increase resource use and energy security. SG 2.0 emphasizes meeting local energy requirements by using abundant resources (solar and wind energy). This reduces long-distance power transmission losses and improves the quality of electricity for consumers.

#### 2.2.2. SG 2.0 Grid Architecture

Four layers make up the SG 2.0′s network architecture:*1.* *Phys Comp Layer;**2.* *Control layer;**3.* *Application layer;**4.* *Analysed data layer.*

Figure 3 depicts the primary layers. Sensory devices in the hardware layer collect data for real-time monitoring. IoT devices, including Smart Generation Technologies, Smart Loads, Smart Sensors, Smart Meters, Phase Measurement Units (PMU), Remote Terminal Units. (RTU, CT, VT) (VT), etc. WSN transmits sensor data. SG 2.0 applications involving energy financial transactions need data.

### 2.3. SG Emerging Issues

Given all the improvements, approaches and procedures in SGs, this emerging technology poses a few concerns. SGs require a crucial SG security technique (hardware, software, infrastructure, utilities, networks, sensors and devices). This subsection reveals the critical concerns in SGs’ communications and information technologies, sensors, estimations, automation system technologies, electrical and electronic devices and energy storage systems. SG’s concerns include the following emerging challenges:

#### 2.3.1. Emerging Issues in Power Electronics and Energy Storage Technologies

Electronic control systems introduce harmonic distortion into the grid and create voltage distortion issues. Indeed, the widespread use of electronic control interfaces (such as flexible AC transmission and high voltage DC installations) will be required to create smart electric power grids [55].

#### 2.3.2. Emerging Issues in Automation Sensing Technologies

Smart meters are attractive automated energy systems which indicate in real time, with two-way access and remote terminal units/interruption, energy consumption, price information and dynamic prices. All components and devices in the smart meter system require additional identification numbers, which makes it more difficult to integrate new devices, appliances, sensors, etc., with an increasing number of customers [56].

#### 2.3.3. Emerging Issues in Communication Technologies

Smart grid communication systems need smart meters and edge sensors to communicate between appliances and the database. Smart meters include a modular, interoperable, reliable, scalable and efficient two-way communication backbone that requires long duration and high frequency [57]. The transmission and storage of information should be protected to prevent cyber-attacks [58].

### 2.4. Energy Prosumer in SG

#### 2.4.1. Energy Prosumer

In [59], the authors state that photovoltaic power plants, solar thermal power plants and trade winds are the most promising energy sources. SGs can enhance the energy consumption, management, distribution, efficiency and use of renewable sources. Some energy rules aim to make renewable energy more cost-effective; however, it is difficult to estimate the costs of adopting renewable energy sources. Energy consumption is evaluated using the LCOE and ESA cost-evaluation methods [60]. The SG system contains many components for power distribution. This consists of intelligent technology, digital networks, two-way communication, integrated management approaches and requirements and regulations. The first systematic research to use SGs based on customer comparisons was [61]. In [62], the authors report that customers are using an average of 100 MWh of solar energy with rising energy prices. By using an energy storage device, energy disturbances can be mitigated. In [63], the authors reported that using edge technology, it is proposed to convert traditional power systems into intelligent ones. Five distinct roles of the consumer in the smart grid are examined.

Market Participation Strategy (MPS), Strategic Analysis, Competitive Advantage, Evaluation of Economic Benefits and Business Research. The consumer can contribute to the renewable energy trade [64]. In [65], the authors explain that energy companies are integrating consumer capabilities to fit consumer demand by combining information technology and lowering prices.

Community Consumer Groups (CPGs) are comparable to energy distribution and management customers. Consumers want to regulate energy production, hours of consumption and storage capacity [66]. Publications and analyses helped us identify the consumer segments of smart grids (see Figure 3). Several specialists have explored leading competitors, innovative grid market tactics and consumer innovation. Several studies refuted the Prosumer concept [67]. One study showed that improved customer collaboration enhanced energy product offerings and reduced the risk of economic losses [68]. To replicate customer behaviors, fundamental environmental factors were used. The program generates energy alliances based on a spatial correlation structure [69]. The results indicate that partnerships reduce network storage and capacity. SG companies are curious about the blockchain and Big Data. In [70], the authors cover the SG blockchain robustness system. The proposed system uses the blockchain and smart contracts to reduce costs, speed up transactions and protect user information. A big data approach is a great way to manage study group data. Figure 4 shows the ProSG PEC and PMC (PMC) classes. The PMC literature examines the market structure, roles, objectives, alliances, incentives, and management. PEC provides economic, technological, social, communicative, evolution and participatory issues. Most researchers have focused on Prosumer management and energy exchange strategies, partnerships, expectations and incentive programs. They highlight the need for new innovative approaches to address these challenges more efficiently (Table 2). Prosumer management consists of the following aspects:Communication/negotiation: for approval and common consensus among beneficiaries;A normative/ethical policy: to maintain responsibility for energy share distributions;Assessment of prosperity: for influencers and influential actors who do not meet expectations.

#### 2.4.2. Suppliers Impacts

The interaction between suppliers and Prosumers is crucial in innovative grid architecture because it affects power generation and supply and demand. It will also be appropriately organized to ensure all parties’ willingness to cooperate, so energy exchange is long-term [71]. In [72], the authors illustrate that prosumers have acquired attention as energy suppliers and customers. In [73], the authors mentioned that the energy market infrastructure allows customers to become suppliers and build ties with other enterprises. The energy market infrastructure encourages prosumers to increase flexibility, competitiveness in the energy business, advanced systems and equipment regulation, economic benefits, economic rewards, low energy costs, and transparency [74]. Prosumer interaction promotes customer preferences and benefit-seeking goals [75]. The accessibility and legislation of emerging technologies, sustainability advantages, financial gain, data and energy consumption are prosumer goals. Effective communication methods can order energy demand by raising public understanding of innovative grid benefits and managing customer trust [76]. Prosumer interaction improves consumer priorities and helps facilities achieve market goals, according to [77]. It encourages users to adopt evolving technologies by presenting sophisticated innovations, environmental benefits, financial prospects, fee figures, energy use, and security data. In [78], the authors suggest evaluating intelligent grids from a social, economic, and technological perspective.

**Table 2 sensors-22-08397-t002:** Prosumer SG 2.0 agent goals.

Prosumer Smart Grid (ProSG) Taxonomy	Prosumer Engagement Class (PEC) Concepts	Prosumer Management Class (PMC) Concepts	ProSG Concept: Related Works	Refs.
**Prosumer Market Design (ProM)**	***	It is characterized by consumers who provide services to the network and turn into active consumers. Depends on the integration of the consumer product network, peer-to-peer models and consumer social groups.	The authors focused on a survey to promote modern technological developments and aim to inspire awareness of further liberalization of the electricity market, especially in areas closer to consumers. A Prosumers agent was presented and explained to consume and produce energy	[76,77]
**Prosumer Alliances (ProA)**	***	Ensures more power to the grid with less diversity, thus using less storage and wasting less energy	A consumer alliance is included to analyze accurate weather data from specific area stations to simulate each consumer’s actual production and consumption patterns. ProA agents aim to collect the required data according to advanced algorithms.	[78,79]
**Prosumer Engagement (ProE)**	Enables consumers to transform into active consumers and build strong relationships with other entities in the network	***	ProE is included in SGs as a variety of electrical resources to engage large power plants, renewable energy systems, energy conservation, reaction needs and electric vehicles. The obtained results show that by aligning with many SG goals, ProE will take a “stronger” position in future energy markets. The actual launching of SGs depends on how customers accept SG services.	[80]
**Prosumer Social, Economic and Technological (ProSET) aspects**	Consumer behavior is affected by ProSET.It seeks to establish exchanges and attitudes in the energy field regarding the value and influence of other households.	***	To achieve greater acceptance of a ProSG for marketers, economic and social environment analyses are necessary. The social perspective of future research is also an important aspect. It may require a specific area of business service, safety, policy and job initiatives. The launch of emerging technologies alone does not promise consumer acceptance, except in the case of accelerated technological growth because consumers find these to come at premium prices.	[81]
**Prosumer Management (ProM)**	***	ProM aims to produce and share surplus power with the grid and other Prosumers.	ProM agents were chosen as essential partners in the future because of their vital role in managing peak demand. Moreover, during power management, it is crucial to test ProM behavior patterns. All the variables that govern peer activity and relationships within the smart grid in identifying grid demand features and energy needs expectations were examined.	[82]
**Prosumer Goals and Motivations (ProGM)**	***	ProGMs have two main goals: scheduling offline required electricity (in advance), and the expected energy consumption is performed in real-time	ProGMs must consider the supply and use of uncensored consumer ability to prepare for expected energy use. ProGMs aim to change the energy structure and enable the consumer community to consider the economic, environmental and living standards of each consumer.	[83]

***: This topic has been cited in this/these reference(s).

### 2.5. Energy Domain Prosumer Classifications

The intelligent city energy sector seems to work, as do energy engineers. Because of their importance in solving the energy consumption problem, energy consumers are an attractive research topic (See Figure 5). Assembly lines, renewable energy plants and energy storage systems (ESS) consume energy [79]. In [80], the authors explain that consumers use SG 2.0 and solar and wind energy to produce and use electricity. Peak-hour consumers use external (grid) power. Excess electricity can be supplied by the self-production of coal or local oil markets. In [81], the authors define each energy consumer as a production-oriented or consumer-oriented consumer (ProCO). The resources of smart cities are depicted in Table 3 [82]:Energy Generation Company (EGC): Commercial energy generators generally collect electricity from existing energy generators and market them to local energy consumption entities. The power company buys energy from separate power plants.Home Energy Storage System (HESS): Energy storage devices for households that own and use reusable energy on demand obtained from small renewable energy plants such as solar or wind power plants.Building Energy Storage Systems (BESS) are designed for energy saving, storage and utilization (renewable energy plants).Electric Vehicle Charging Stations (EVCS): This infrastructure requires power sources, batteries and computer networks for charging. The electric vehicle’s charging station collects energy from the power source and sells it to the car, storing the point in the car via the battery. It also acts as an intermediary in selling electricity between electric vehicles, homes, networks, etc.Green Electric Vehicles (GEV): EVs only include green vehicles with a battery and an electric motor inside the car that is capable of transforming velocity into electricity and maintaining it in the artillery during service. The electric vehicle markets power through the charging point.Prosumer Smart Home (PSH) without ESS: The PSH is provided only with the device or related components that can control the power required and consumed by the network without installing and considering ESS.Prosumer Smart Buildings (PSB) without Building Storage System (BSS): PSBs are limited to similar systems or devices that can manage power supplies from the power grid.Solar Energy Company (SEC): Energy companies produce a large proportion of renewable energy. Residential solar companies have a capacity of more than 1 MW. The power supply is calculated by the Prices of Electricity Sold (PES) for typical and electrical generators.Wind Power Generation Company (WPGC): Wind farms provide wind energy. Domestic wind turbine production is about 1031 MW. The energy supply is calculated by the prices of electricity sold (PES) for typical and electrical generators.BSS with Large Capacity (BSSLC): BBSLC stores and uses renewable energy to increase economic productivity. According to some statistics, many companies have installed storage solutions with a capacity of more than 1 MW installed, with an agreement strength of about 20 MW each year [83].

## 3. Peer-to-Peer Energy Trading

### 3.1. Prosumer and Consumer Cases

The resources of P2P energy exchange are a peer-to-peer sharing system in which renewable energy consumers and small cooperatives distribute electricity to residences, businesses, etc. P2P Energy Exchange is a peer-to-peer service in which renewable energy consumers and small cooperatives provide electricity to homes, businesses, etc. (Appendix A). P2P technology permits novel energy models [84]. According to [85], the change in electricity distribution technologies and trends will cause energy prices to adapt to a competitive and automated economy. Emerging in the energy sector is peer-to-peer power generation. According to [86], P2P enables consumers to become producers and exchange surplus resources with competitors. On-site PV self-consumption is utilized. Energy storage increases individual usage. According to the findings, intermittent generation of renewable energy results in uncoordinated power to and from the system. Customers cannot be awarded or punished by utility networks. In [87], the authors highlighted that the low need for self-regulation is the most crucial element for customers in intelligent cities.

In [88], the authors list and categorize the basic components and technology involved in P2P power exchange:Consumer Power System Level (GPP) includes power lines, transformers, smart meters and charts. These units are the infrastructure for P2P energy trading.The Consumer Management Level (PML) manages the power grid. This layer regulates energy flows using efficient energy sources. Voltage and frequency control are examples of control system control duties.Consumer Business Level (PBL): counterparties and private companies. This includes electrical investors, manufacturers, DSOs and regulators. In this layer, many P2P power trading strategies can be incorporated.

### 3.2. P2P Energy Trading

SGs require energy trading technologies due to the shortcomings of traditional scattered energy trading and the suggested blockchain-based model (i.e., infrastructure-based P2P energy trading). Local microgrids and power suppliers will utilize a blockchain network to implement the mentioned P2P energy-sharing idea (Figure 6). Customers can import from another consumer or purchase from conventional power sources [89]. Since the blockchain is unchangeable and distributed, this architecture guarantees that all transactions are accessible to customers, big energy providers and governments. Government supervision of the energy-sharing industry requires a forum [90]. All parties will have increased vigor and prospects. This agreement will afford traditional suppliers new business prospects by compensating distributors for these facilities. This design has intelligent components on both the consumer and power supply sides. There are four levels. Large generators and energy distributors comprise layered power systems [91]. During data transfer, pre-negotiation and conversation will take place. The transaction consists of three phases. Initially, the customer needs energy. Buyers select among sellers’ offers. Also essential is the communication between vendors and the network. The vendor is required to consent to the dissemination of energy. The distributed ledger stores all transactions [92].

### 3.3. Infrastructure-Based Energy Trading

Traditional electricity transactions are centralized. P2P transactions do not, however, require a central authority. Energy can be shared directly between consumers, assuming its physical conveyance [93]. Two adjacent housings can transmit power via a wired connection. In infrastructure-based P2P energy sharing, users have smart meters and Internet of Things (IoT) sensors installed on the device for which they are purchasing energy (e.g., Home-To-Vehicle-V2H) [94,95]. Figure 7 illustrates how devices connect across the blockchain to facilitate efficient transactions. Consumers expect that resource exchange is possible. Consumers engage in commerce with other consumers. Without intermediaries, consumers can perform transactions if they can transport electricity. As the talks occur on a different network, the connection to the blockchain is severed [96]. The Brooklyn micro-grid is an example of infrastructure-based P2P. It depends on particular customers and connected customers (five) [97]. Using intelligent meters and e-wallets, consumers can sell excess electricity to their neighbors. Each participant has access to all transactions capable of self-execution. Users can choose the total cost and energy type (i.e., conventional or renewable energy). There may be operational issues if the network is disrupted prematurely. One is the provision of physical infrastructure for world-scale energy conversion [98].

### 3.4. Smart Contracts in Energy Sector Applications

#### 3.4.1. Smart Contacts

Smart contracts are blockchain applications that run when standards are met. They are used to automate an agreement’s implementation so that the outcome can be guaranteed to all parties without intermediary intervention or loss of time. They can also automate the procedure to start the next activity [99]. By solving future SG 2.0 issues, blockchain’s architecture, which is irreversible, transparent, secure, auditable and robust, has opened up new opportunities in the energy sector. DLT is an effective technology for decentralizing operations while ensuring data accuracy and reducing third-party influence. This eliminates central transaction costs and the possibility of failure at a single point. Bitcoin blockchains are implemented. Blockchain’s essential characteristics are quickly spread to the energy, agriculture, banking and healthcare industries. Transaction blocks are arranged in chronological order on the blockchain. This ledger documents all past transactions and is shared across peers, known as nodes, to ensure transparency. Each block has a cryptographic hash associated with it. This one-way mathematical function generates a unique output from the input data, which is a block reference. Changing the input data changes the hash algorithm, making the blockchain immutable and secure. Blockchains enable stakeholders to distribute responsibility without third-party supervision. In addition to PKI, the blockchain allows user authentication and data privacy [100].

#### 3.4.2. Energy Sector Applications

Applications of Smart Contracts for Energy. After discussing the primary characteristics and advantages of smart contracts and the methodology and actions required to execute them, we examine their claimed energy consumption. Two innovative contract applications are left-side control and flexibility and right-side distributed control and flexibility. The most apparent application is trade and payment because intelligent contracts operate on a blockchain created for financial transactions. Typically, smart contracts are utilized in energy or flexibility trading studies. In these applications, smart contracts facilitate the matching of consumers (offering microgeneration and/or storage) and recommend a safe payment or settlement mechanism [101].

In [102], the authors developed a P2P power trading system using Ethereum smart contracts. The smart contract resides on a shared blockchain, ensuring accurate transaction execution and immutable transaction records. It eliminates the high costs and overheads of typical server-based P2P power transmission systems. Dynamic pricing for automatic balancing of aggregate supply and aggregate demand within a small network, double selling prevention, intuitive and independent running, test base experience (Node.js and web3.js API to access Ethereum Virtual Machine on Raspberry Pi with MATLAB interface) and simulation via characters are the main features for implementation (default consumers and persistent consumers of the standard). In the following, we describe the primary state schemes and implementation methods.

In [103], the authors mention that smart contracts serve as the basis for distributed blockchain applications. The authors highlight that intelligent contract adaptation is not available, and source code reuse is limited to cloning. For this reason, the authors discuss the architecture and implementation of the smart contract with precise validation rules. First, update the list of smart contract validation criteria at runtime to set the distinct transaction types. Second, it is possible to reuse validation rules across smart contract configurations. Using UML models that are independent of the blockchain, smart contracts and validation rules are developed. Java was used to implement the pattern. The design controls are inherited through polymorphism and sealed classes, assigning them only to final classes. This approach facilitates the reuse of smart contracts and security. Due to the reusability of the test class between smart contract configurations, the announced validation requirements improve test automation and reduce test preparation effort. The exchange of renewable energy within the consumer group and between cultures refers to the pattern of consumption.

In [104], the authors note that microscopic renewable energy production is rising, resulting in consumption patterns in society that allow for energy surplus and consumption flexibility. Peer-to-peer energy trading requires an immutable decentralized system, and access is controlled for token energy assets. Consumers, electric vehicles, energy companies and storage providers can use a unified blockchain-based system to trade energy assets. Hyperledger Fabric supports two versions of the system. Non-fungible (NFT) tokens encapsulate a unique identifier or information and value, while fungible tokens represent value (FT) only. We have created and tested ways to manage the token lifecycle in smart contracts.

In [105], the authors state that demand response (DR) services can enhance renewable energy penetration by controlling load usage and system balancing. The success of industries, societies and consumers that provide and integrate load flexibility into energy markets will depend on redesigning and adapting existing stakeholder relationships. With the increasing contribution of smaller assets to resilience, new challenges will arise, such as transmission coordination, DR delivery validation and contract settlement, while ensuring secure access to data. The authors used distributed ledger technology (DLT)/blockchain to securely track DR provisioning, emphasizing validation, data integrity, source, rapid logging and sharing within an authorized system across all necessary stakeholders (including TSOs, pools, etc.), DSOs, BRPs and consumers. They designed and built the DR framework as a proof of concept on Hyperledger Fabric, using tangible assets in a lab environment to assess its applicability and performance. The lab is equipped with a 450 kW energy storage unit that provides DR services on demand from the system operator and is published on the blockchain. When there are fewer than 32 requests per second, the total execution time is less than 1 s. The memory usage of smart contracts did not exceed 1% for both active and passive nodes, while peer CPU usage remained below 5% across all simulations. The CPU consumption of smart contracts has remained below 1%. Scalable implementation results allow DLT to be deployed in the real world to facilitate the development of flexible markets using blockchain technology.

In [106], the authors say government subsidies enable more families to install renewable energy sources, such as photovoltaic (PV) panels, to become grid-independent and save money. Consumers can sell excess electricity to the grid only at regulated prices in Romania. Allowing consumers to sell the surplus back into the local network can increase their number. Consumers will pay less for power that is not subject to grid fees, which leads to lower prices. Peer-to-peer contracts signed at the micro (G) level can facilitate this exchange. The authors introduced a new trading platform based on P2P smart contracts to trade surplus power to consumers in a micro-local network. Several trading scenarios are shown, which allow trading depending on the locations of the participants, instant active energy demand, maximum daily energy consumption and on a first come, first served basis in the blockchain-based trading ledger. The Small Grid, Low Voltage (LV) model is used to test the technology intended for real-world deployment. We compare the transferred amounts and financial rewards for different scenarios.

## 4. Blockchain Application in SG 2.0

### 4.1. Motivations of Applying Blockchain in SG 2.0

The SG is an evolving approach that integrates network computing and digital technology to modify and modernize the network legacy in power distribution and create a more reliable, efficient and perceptible transmission network [107]. The demand for renewable energy sources and the severity of climate change prompted these modernization adjustments. The main objective of these updates and transformations is to change the current energy environment by integrating dispersed energy supply with sustainability and use and by reducing dependence on fossil-fuel-based generations. While the new network model brings producers and consumers closer together by deploying renewable sources as independent distributors, the old traditional network serves customers with long-distance transmission lines [108]. Although the central structure of the current architecture is one of the main problems, the smart grid and the Energy Internet are designed to adapt to distributed and centralized energy generations. Markets, transmission and distribution networks and power generation depend on primary or intermediate institutions to track, receive, process and assist all aspects with appropriate control signals in this centralized environment. These key organizations can follow, receive and coordinate data linked by innovative grid elements. In addition, the energy demand is usually sent over a long-range network to transmit power to end users through the distribution network [109]. The most recent architecture for innovative grid systems creates questions in light of the increasing proportion of renewable energy and the number of components, scalability, high computation, connection pressures, availability fits and the difficulty of managing multiple-component power systems [110]. Decentralized infrastructure enables network operations that are more complicated, intelligent and proactive. The design is advancing toward a fully integrated network with aggregated configurations to maximize complex interactions among all new network components. Synchronization of EI and its use contributes to the most cost-effective, efficient and trustworthy new network service [111].

The energy market for the issue is expanding excessively. SG ensures efficient power transfer with low losses and maximum efficacy. Individuals can supply the grid with the least quantity of energy required. This approach complicates current infrastructure, such as the handling, evaluating and recording of transactions between generators and consumers [112]. This section demonstrates how blockchain technology can process network transactions. The network checks transactions involving intelligent contracts. The blockchain guarantees the integrity of transactions between consumers and producers. It provides a consistent basis in marketing, aiding audits or conflict settlement [113].

### 4.2. The Evolution and Structure of the Blockchain

In the past two decades, blockchain technology has grown significantly, from Bitcoin (blockchain 1.0) to Ethereum (blockchain 2.0) to killer categories such as cryptocurrencies (blockchain 3.0) (See Figure 8). The system has transitioned from a simple database to a network of cloud storage [114]. Ethereum’s blockchain capabilities transform it from a database-only cryptocurrency service into a public infrastructure capable of supporting several decentralized applications in financial services and other sectors that could profit from digital currencies. The challenges faced by the first and second generations of blockchains hindered their acceptance [115]. Using the consensus technique, proving ownership of an object without a central authority requires considerable time. To execute a transaction on the Ethereum blockchain, each node must calculate all smart contracts in real time, which is time-consuming. A blockchain consists of time-ordered, encrypted blocks connected by a hashing algorithm. Blocks with temporal names are immutable—a coalition of coefficients [116]. Bitcoin transactions are monetary transfers. In our instance, power stands in its place. The result of the hashing algorithm is stable and independent of the input. In [117], the authors stated that a minor modification to the information can alter production. It is simple to compute the exit based on the entry, but not otherwise. The block is punctured with H(x), where x is the block number when it is complete or ready to form a new partnership. To develop a “string”, the hashtag is saved in the following block. Before the last block, the function was iterated so that the wrong change may be swiftly alerted. Bitcoin can transfer funds and change the currency’s owner. Prior and succeeding owners can be identified by their addresses [118]. The public and private keys produce the speech. The network promptly validates the property owner’s health. Double spending is eliminated because the purchase history is recorded on the blockchain.

### 4.3. Blockchain P2P Energy Trade

Prosumers and SGs generate new options for electricity exchange for participants (including consumers, grids and energy storage). This paradigm shift in energy trading promotes energy economics by establishing a reliable structure [119]. Trading processes must be decentralized to increase the number of market participants (as shown in Figure 9). In [120], the authors reported that blockchain can create a private, decentralized, robust and stable platform for electricity trade. The leadership of blockchain enables transparency and a distributed chain to manage verified transactions properly. Since the database was shared, blockchain has a clear consensus. The block size is the number of transactions contained in each block. The blockchain comprises the network, agreement, storage, visibility and side planes. The network level is responsible for the connection, whereas the storage level manages the blockchain. Consensus is crucial because it ensures a modern and inclusive network [121]. There are three sorts of blockchain players:Validators solve a cryptographic problem to provide their authenticity (through mining).Partial nodes do not participate in authentication but back up the network registry.Users generate transfer and contact visuals for data mining. Mining results in the creation of chain blocks.

Blocks are comprised of transaction information and hash values [122]. Consumers/Prosumers and SG 2.0 provide participants with energy trading choices (including consumers, grids and energy storage). This paradigm shift in energy trade improves energy economics since energy is the most powerful tool for economic progress. Trading systems should be decentralized so that more people can participate securely. Blockchain can develop a robust, distributed private energy trading platform [123].

Blockchain is a potential peer-to-peer energy sharing technology, but it faces various limitations [120]:Blockchain is scalable, reliable and secure;Blockchain’s development cost is a big drawback;A transaction validator takes a lot of processing and internal training, which increases the cost of a typical database system;Blockchain is showing promise in previously changing electrical grids.

After each account, blockchain transactions are added. This process is complex and time-consuming. The different objectives define the technological, organizational and economic architecture of blockchain-based services. Technical specifications aim to:i.Scalability: modify models to add new triggers;ii.Develop models of energy exchange without a central authority;iii.Manage Sensors. There are different sensors:a.Two cars or two homes can exchange electricity;b.Models must include many devices and technologies to support many types;c.Two benefits of intelligence: first, electricity can be delivered inexpensively;d.The consumer must choose from among the bids submitted;e.Internet of Things (IoT) computers power decentralized blockchains;f.Electric cars contain IoT devices and communications sensors.


### 4.4. Role of Blockchain in SG 2.0

Decentralized power transmissions in SG 2.0 offer prospects for the blockchain. Local micro-networks allow P2P power to be traded between consumers without repudiation [121]. Each node will keep a copy of all transactions for all participants, which is constantly updated. The logbook provides user authentication and data integrity in [123]. Smart contracts can be used for P2P energy trading, autonomous vehicle charging, miniature grid energy trading, seamless integration of renewable energy networks, network automation and distribution network management [122]. Enabling the next level of energy markets, blockchain provides consumers with discretion, security, transparency, refutation and independence. Blockchain technology encourages the integration of complex computing resources into distributed processing, thereby enhancing trust between the end nodes and the network operator. Blockchain will protect power data management and private cloud storage [119].

### 4.5. Transaction Workflow

The workflow is divided into three sections. Any communications or negotiations between the buyer and seller are referred to as “energy deals” [124]. One example of interaction before a trade is the posting of a user’s offer/request over the network. Various mechanisms can be used to protect data and confidentiality (See Figure 10). The seller’s bids are recognized in the bids between the buyer and seller to determine the user’s right to be treated with respect [125].

Entities: engage in energy trading and are categorized into three types. These companies will be able to use smart meters and blockchain technology.The central utilities manager consists of the community, electricity providers and network owners who own the physical and technological resources for energy sharing and transmission. It is responsible for formalizing global order and regulation.Energy generators: those that have large reserves of energy supply the grid with energy that uses conventional and renewable energy sources. Examples of commercial energy suppliers include national grid monitors, small grid owners and turbine owners.Energy to consumers/prosumers: Consumers have the added benefit of producing and distributing excess electricity to other network consumers. Private homes, hybrid vehicles and large structures can be consumers/prosumers.

Acceptance of payment methods is one way to maximize consumer value. As a result, the consumer will be more excited because they will receive instant rewards and investment opportunities. Second, there may be a way to encourage repeat customers. For example, if the consumer sells to the government, the buyer can be paid with power, cryptocurrency or changes in their bill.

In peer-to-peer energy trading, real-time monitoring and supervision is critical. The idea behind demand response is to shift the energy burden from low-demand users to high-demand customers. For example, a household requires less energy in the morning than an office building does. Likewise, the situation reverses at night. As a result, power can be distributed as needed.

### 4.6. Prosumer Energy Management Algorithms

One of the important and exciting features of the SG is the efficient use of the power system features. Various optimization techniques are used for prosumer-based energy management and smart grid features [126]. One of the SG’s important and exciting features is the effective use of power system technologies. For prosumer-based energy management and SG applications, various optimization techniques are used. For example, the authors reported that to achieve streamlined results and improved usage, costs and satisfaction of all stakeholders, Prosumer Energy Management (PEM) should have relied heavily on optimization algorithms [127]. Some descriptions of different modelling methods for energy conservation and PEM optimization algorithms are discussed, as shown in Table 4.

#### 4.6.1. Prosumer Genetic Algorithm

Genetic algorithms are considered an attractive research issue due to their ability to solve the DSM and complicated economics. Furthermore, PGA is applied to increase energy transfer efficiently at a calculated level. GA has been involved in many areas of the energy system to solve optimization problems. The goal of the schedule should be to meet all plant and system constraints while meeting the demand for load at the lowest operating cost. Genetic engineering and development focus on PGA improvement strategies. In [129], the authors presented the PGA as a potential solution due to its ability to overcome the complex problem of improvement and is efficiently employed in various fields and sectors. The authors stated that the diversity of problem formulation is one of the essential advantages of PGA compared to other optimization techniques such as linear optimization or dynamic programming. This means that the PGA can deal with different types of restrictions. First of all, the strength of each Decentralized Generator (DG) must be kept within its range and handle various types of prosumer energy management concerns. In [133], the authors proposed PGA-controlled and supervised in real-time Decentralized Generators (DGs) and load transmissions based on models and time constraints associated with start time rather than optimum efficiency. A GA-based approach was proposed to control energy demand. The load usage is governed by taking into account the set point: the load that the consumer wants to operate within the cover of the set point. This proposed approach aims to control the use of the pack by observing a specific issue, that is, the prosumer payload is willing to work within the maximum specified point.

#### 4.6.2. Prosumer Mixed-Integer Linear Programming

For the design of high-dimensional and non-linear systems, the PMILP was proposed. The classic principle of operation is applied to accelerate the cycle towards the device’s expandability. Indeed, PMILP is a methodology used to automate storing electricity in an intelligent system. PMILP differs from other dimensional methods of programming, which contain both actual and incorrect variables. The PMILP was introduced for designing high-dimensional and non-linear systems. The classic operating concept is applied to accelerate the process toward the expandability of the device. PMILP is a method used to optimize the handling of energy in a smart network. PMILP differs from other dimensional programming methods, which deploy actual and incorrect variables [129].

Many innovations usually generate just one ideal solution, while others may create many solutions. In [133], energy-efficiency algorithms focused on the recommended PMILP were proposed to reduce costs and conserve electricity. Indeed, the authors proposed the PMILP-based energy efficiency algorithms to reduce costs and save electricity.

The results showed that the total cost of the optimization problem related to energy consumption was reduced through optimization techniques. The PMILP algorithm was used to encourage average users or residents to change purchasing costs to keep costs down. The authors explored and measured the concept of time limits.

#### 4.6.3. Prosumer Particle Swarm Optimization

PPSO has seen growing numbers of applications for SG domains; the three largest application areas are scheduling, active control and network layout scheduling. Indeed, SGs are used in PSO variants, including genetic SPO, unexpected PSO and quantitative PSO. PPSO is often used in the smart grid to control electricity. A Particle Swarm Optimization (PSO) algorithm was proposed in [136] to minimize energy expenses for economic transmission issues related to demand exchange and for a random process that begins to create a series of alternatives. Indeed, PSO was also presented to reduce operating costs and energy efficiency in conjunction with natural gas networks. They deployed PSO to implement a natural gas generator for the small grid to address problems in clean energy supplies and reduce the load and congestion of the gas. To prevent pollution and balance payments, the efficient distribution of all digital networks must be synchronized with the electricity grid. A transition is made to use PSO to address the related network issue.

In [134], PSO has been shown to outperform some standard methods used based on Information Engine Services (IES) and based on operating expenses appropriate for IES. The authors included challenging PSO improvements that converge faster and require less computational time. In [135], the authors discussed the PSO’s improvements, such as the fast convergence in less computational time. In [136], PSO was applied to obtain the optimum energy flow for renewable energy wires.

#### 4.6.4. Prosumer Linear Programming

For storing electricity for the smart grid, the PLP optimization algorithm was selected as an appealing design approach [136]. In [137], the authors claim that PLP was applied as a linear function of choice factors to determine the best strategy for solving objective function issues and restrictions. Numerous scientists have employed PLP in different energy storage devices, according to the authors’ report in [138]. To boost daily consumption from peak demand, linear programming (LP) is applied. In [139], the authors used the LP as a consumption schedule to reduce peak demand at home. The authors also proposed a linked network that links the grid, house, power plants and integrated power management system. In [140], the authors suggested an LP strategy to increase energy needs by utilizing local green energy resources. By utilizing the LP and the power grid, the authors state that some have attempted to illustrate the distinction between energy production and end usage. In [141], the authors mention that the proposed power grid concept makes sustainable use and the use of municipal solid waste.

#### 4.6.5. Prosumer Integer Linear Programming

The PILP is an additional SG 2.0 technique for enhancing electrical infrastructure. Only numerical and binary values can be utilized, which makes PILP different from LP. ILP can be used to convey additional inquiries, just as LP can. The functional area of the LP model is defined by the constant restriction of each vector to a single continuous period [142]. The structure is PILP if the variables are constrained to proper values. The ILP model is substantially different from the LP model since the region is realistic. It is critical to remember that these models can be used to explain a variety of LP sub-processes and are pretty comprehensive in the real world as a crucial area for LP programming implementation.

In [143], the authors recommend an ILP-based demand management system to raise the load for end-users in smart grids to account for the electrically moved equipment and the displacement over time. To put more pressure on smart grids for users to move the electrically delivered equipment and the equipment over time, the authors propose a demand management program employing ILP [144]. Smart meters, appliance interfaces and home appliances are the network’s three main building blocks. Smart meters are crucial to the proposed system because they allow interface-based client data collection from devices and usage plans. The smart meter was created utilizing the data optimization method, according to the authors’ statement in [144]. The three main elements of the proposed network are smart meters, appliance interfaces and home appliances. The suggested approach to gathering customer data from gadgets and consumption plans via an application heavily relies on smart meters. The algorithm used for data optimization was used to create the smart meter. The mathematical formula for the order scheduling technique is stated as having a linear maximization characteristic that lowers the daily load.

## 5. Open Issues and Future Directions

Demand response and market management for unexplored areas is still under study, and applications based on machine learning for energy efficiency and cost analysis may include peer-to-peer energy trade. For example, a real-time billing system can optimize energy pricing based on current and potential energy prices (forecast) and charge the consumer accordingly. The blockchain is viewed as black box in most blockchain solutions. For example, many strategies [122,123,135,136] use smart contracts as the blockchain protocol to grow the architecture. This limits the leverage over the overall architecture and performance of optimizations that cannot be made to the blockchain used in smart contracts. In the future, instead of using the blockchain as black boxes, the blockchain could adopt a problem-specific approach to energy trade.

There is a need for a network in which all prototypes can operate as a common framework and adapt their behavior to consumers’ needs. For example, consumers must be able to sell electricity domestically and globally for large-scale energy storage systems.

The traditional architectures of energy supply smart meters and every other revolutionary system are not used in blockchain. Many prosumers/consumers are eager to adopt this architecture. Most of the energy sharing frameworks presented to us presume that prosumers and consumers have intelligent devices. This new architecture blends conventional design with a cryptocurrency. Consolidated energy trade by a group of consumers is outperformed by the inefficiency and robustness of individual consumers operating as autonomous firms in terms of renewable energy supplies. In addition, the power source for individual consumers may be insufficient to handle conventional power generators and may be unpredictable due to climatic conditions.

### 5.1. SG 2.0 Values

The new design must provide more benefits than the alternatives to be useful. SG 2.0 differentiates itself through scalability and manageability. In [122], SG 2.0 is shown to be adaptable and reusable. Smart buildings and network systems can be more easily controlled, managed and coordinated using decentralized problem solving in SG 2.0 [28]. As detailed in the articles, the sparse nature of SG 2.0 and improved construction management processes have resulted in increased energy performance and comfort. Ineffective and unsustainable [48]. Recent controversies have focused on the convenience of the population [33,89]. Giving occupants enough personal control may make them more tolerant and prefer larger settings, resulting in lower power consumption [56]. Giving builders more control and involvement can backfire if they make reckless and ineffective decisions. Users and administrators may have conflicting goals (optimal convenience for users versus optimum operating cost for managers). To maximize build performance, SG 2.0 can control correlations [33,40] (See also Table 5).

### 5.2. SG 2.0 Operational Challenges

The management of construction actions and interactions was the objective of SG 2.0. Decades of research have supported SG 2.0 [12,118,129]. As expected, not many people have embraced it. Based on scalability, reusability and ease of design, developers have difficulty delivering a robust business case for SG 2.0 [15,16]. Open data platforms need DSM to communicate across many build systems and loads for maximum performance. All of these systems must be recognized by SG 2.0, allowing information sharing between agents and devices [105].

Scope implementation issues [12,80,131] are hampering the implementation of SG 2.0 and the demand for an intelligent enterprise-based information system. Demand-management issues arise from squatter tenants, building characteristics, needs, regulatory constraints, power grid dynamics, operational requirements and market constraints. Due to costs, lack of system support, advanced technology and logistical difficulties, these systems are still hypothetical [17,56]. Few studies have examined the interaction of the entire power network for multifactor load control (Table 5). To reduce demand, SG 2.0 hierarchical demand management systems check demand-side resources. Demand management cannot function without a precise connection between supply and demand. Sharing information becomes difficult when there are many vendors, and decisions must be made quickly. In some circumstances, vendor-specific tools are required for separate configurations of multiprotocol gateways or devices [109]. First, a common platform is required for many systems from different vendors. Proxy-based infrastructure is available in many systems [69,104]. The actors in the building system can be closely linked thanks to ontology [59,100] intrinsically. The best ways to generate data include ontology, standardization and data integration. Most research has indicated that communication between agents uses different languages and semantics. One ontology is required to create the systems. Information exchange and decision making from the SG may be inhibited by the lateral volume 2.0 of process-related information [61].

## 6. Conclusions and Future Works

This survey focuses on Prosumer SGs and the main aspects investigated in terms of monitoring and communication capabilities. The current and monolithic technologies of Prosumer SG 2.0 require additional effort to create an autonomous and decentralized intelligence-level concept. The Energy Internet is described in depth and discussed to improve connectivity through the continuous production of Prosumer SGs. Indeed, several unresolved questions and technological obstacles regarding the future of energy management have been identified. In this context, new difficulties will make it possible to develop promising research in industrial and professional fields. Deep knowledge of Prosumer SG 2.0 and its interactions will enable consumers to accurately assess problems/solutions and adopt SG 2.0 innovations, such as blockchain architecture and IoT computing, as per the research hypothesis. Several kinds of research, including smart and public markets, household energy demand from customers and stakeholders and service provider energy consumption, have been identified based on relevant businesses. In addition, we discussed the concept and methodologies used in the literature for energy management based on ProSG. PGA, PMILP, PPSO, PLP and PILP are some essential methods used and evaluated by the writers covered in this survey article.

Furthermore, Prosumer and consumer scenarios for P2P energy trading are described in detail. In contrast, future SG 2.0 energy management systems are shown and detailed. In terms of future improvements, it is highly recommended that consumers and the market pair with blockchain technology to ensure customer efficiency and enhance multidisciplinary electrical home appliances. SG 2.0 and blockchain standards will be enhanced by ongoing research. Blockchain technology and remote management security can be combined efficiently. The long-term vision will provide potential consumers with a better environment with reliable and intelligent mandates and keep consumer costs low.

Future improvements to SG 2.0 technologies will be incorporated to broaden their application range. This relates to creating a network and model for local underground vehicle transportation. To achieve this, a challenging algorithm will be developed and evaluated. Modern technology immediately addresses energy use and cost concerns. Building and installing the entire SG 2.0/IoT platform in a lab environment will be the primary focus of future work during the later phase.

The quantity and nature of the source databases are the first vital limitations of this study, although the selected groups are generally reputable and representative. Second, this sector’s rapid development hampers the survey’s timing. Third, the summary of research activities in many blockchain applications in SG 2.0 may not accurately reflect how they are used or affect people. Based on its findings, this review attempts to determine how the scientific community will respond to current events.

## Figures and Tables

**Figure 1 sensors-22-08397-f001:**
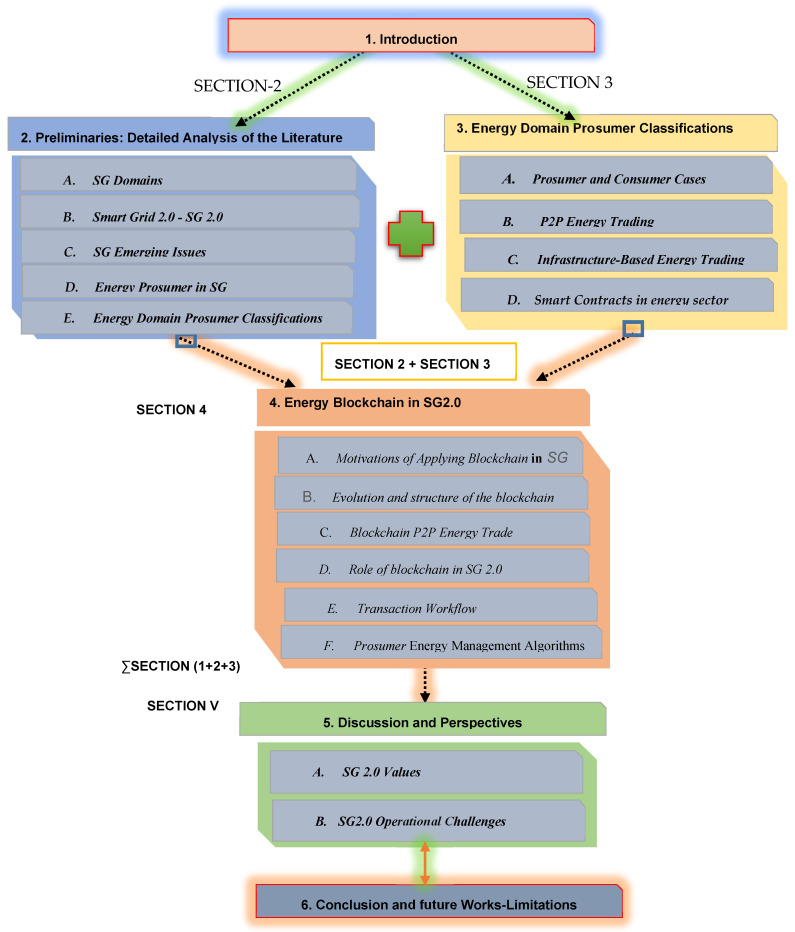
Description of survey paper sections.

**Figure 2 sensors-22-08397-f002:**
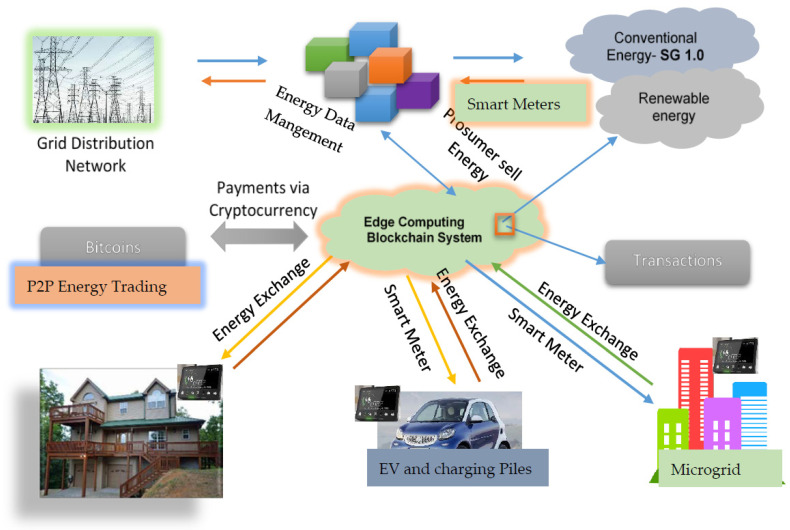
Overall structure of a Distribution Prosumer/SG 2.0 System.

**Figure 3 sensors-22-08397-f003:**
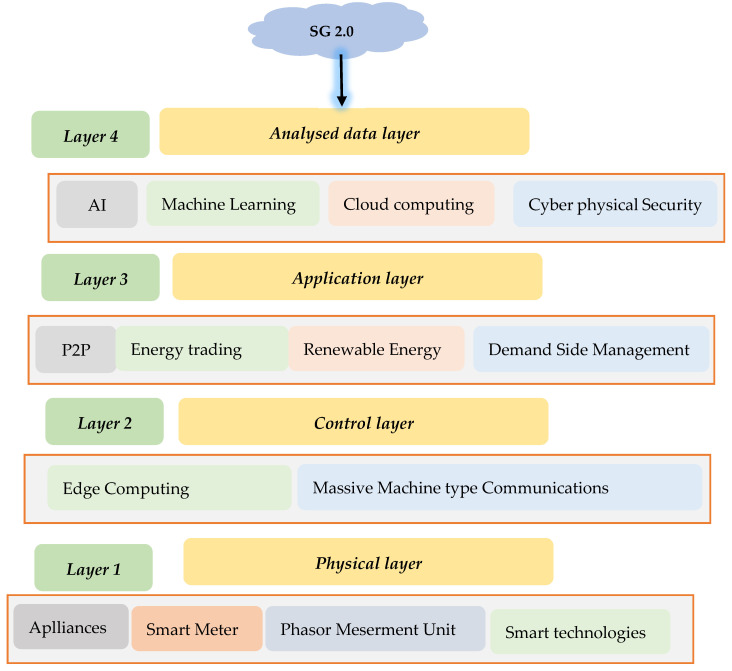
Smart Grid 2.0 Layers.

**Figure 4 sensors-22-08397-f004:**
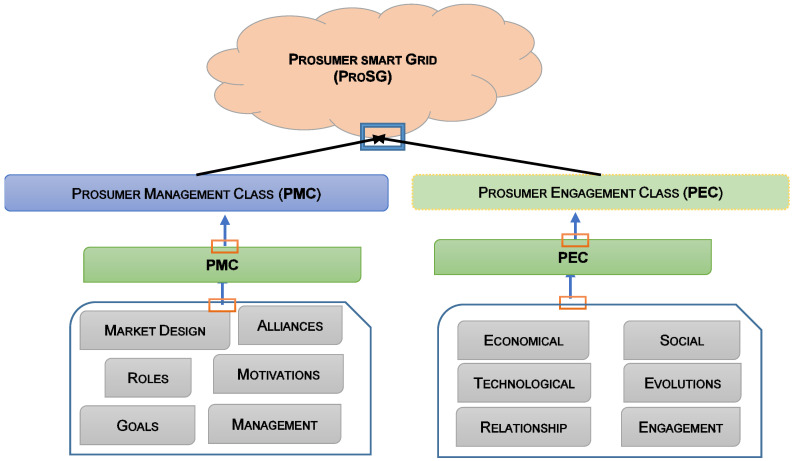
Smart grid Prosumer classes.

**Figure 5 sensors-22-08397-f005:**
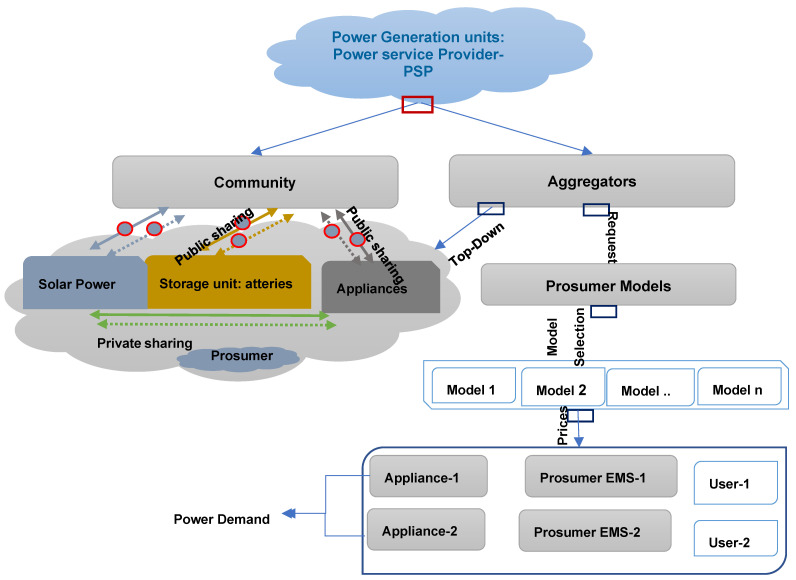
Prosumer Energy Management Scheme.

**Figure 6 sensors-22-08397-f006:**
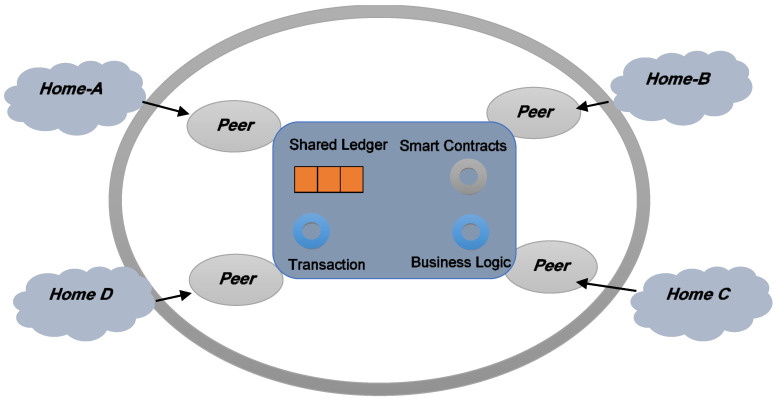
P2P energy trading.

**Figure 7 sensors-22-08397-f007:**
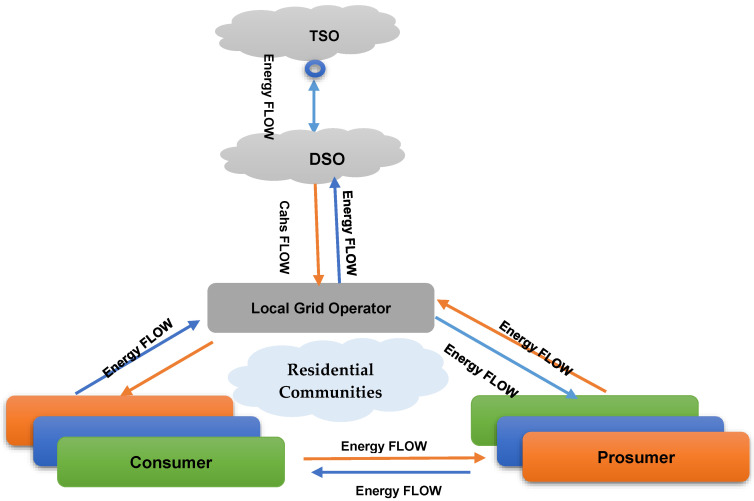
Infrastructure-based energy trading.

**Figure 8 sensors-22-08397-f008:**
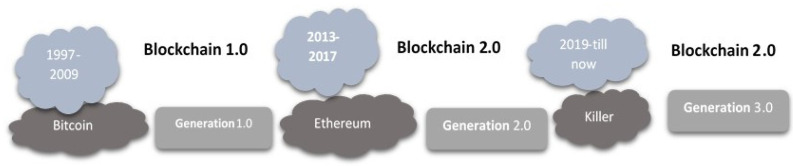
Uprising of Blockchain technology.

**Figure 9 sensors-22-08397-f009:**
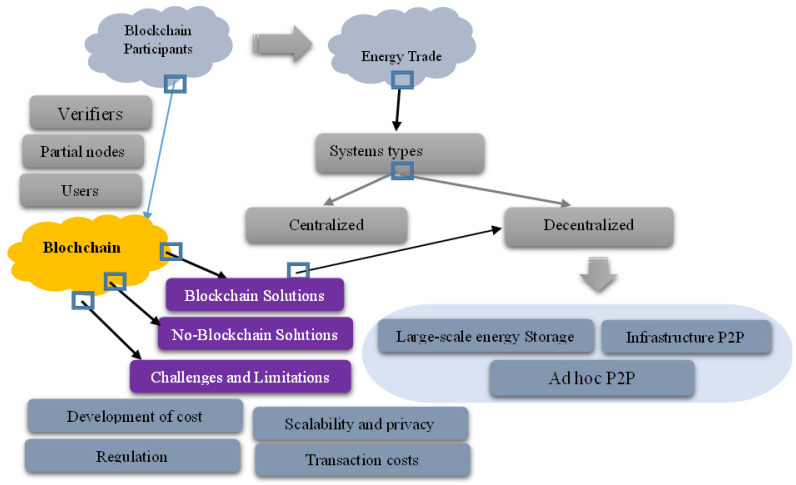
Blockchain P2P energy trading structure.

**Figure 10 sensors-22-08397-f010:**
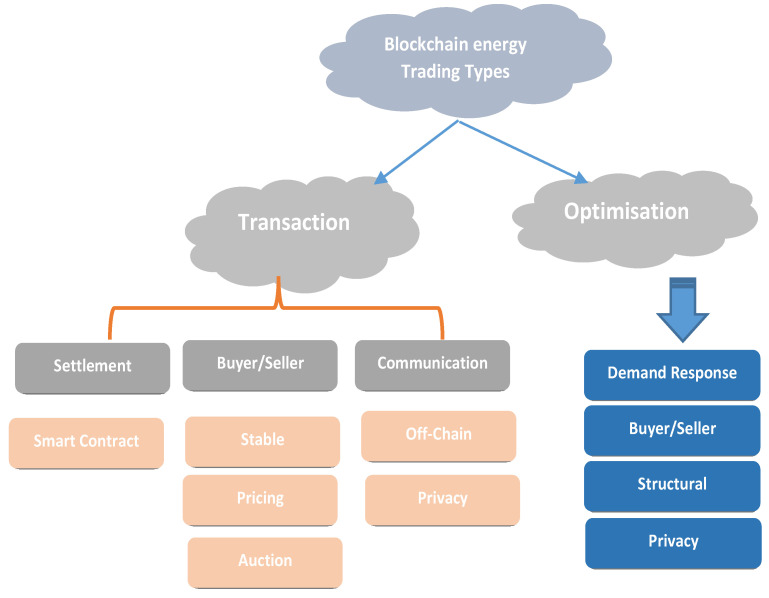
Blockchain-based energy trading Taxonomy.

**Table 1 sensors-22-08397-t001:** Comparison of existing survey papers.

Areas	Concepts	Applications	Field Survey
Prosumer Smart Grid	3	0	3
Prosumer Energy Management	3	3	3
Prosumer Models	0	0	0
Prosumer Concepts	3	3	0
Prosumer Techniques	2	0	0
Economical	0	0	0
Social	0	0	0
Technological	3	3	3
Evaluations	0	0	0
Buildings Clusters	2	0	0
Demand Response	1	3	1
Exchange In SG	3	0	0
Market	0	2	0
Multiple Agents	0	2	0
Energy Trade Concepts	3	3	3
Energy Trade SG Concepts	3	3	3
Blockchain SG Architecture	3	2	1
Edge Computing Methods	3	2	3
Infrastructure	0	0	1
Blockchain Prosumer Architecture	2	0	3
Blockchain SG	2	0	0

3: Non-Use; 2: Low Use; 1: Average Use; 0: High Use.

**Table 3 sensors-22-08397-t003:** Energy prosumer classifications.

Classifications	ProPO	ProCO	Energy Prosumer Categories
WPGC	√	************	Po-Energy Prosumer
HESS	√	√	Energy Prosumer
SEC	√	************	Po-Energy Prosumer
BSSLC	√	√	Energy Prosumer
PSB	************	√	Co-Energy Prosumer
BESS	√	√	Energy Prosumer
PSH	************	√	Co-Energy Prosumer
GEV	√	√	Energy Prosumer
EVCS	√	************	Po-Energy Prosumer
EGC	√	√	Energy Prosumer

************: This topic has been cited in this/these reference(s).

**Table 4 sensors-22-08397-t004:** Comparison of optimization PEM techniques.

Prosumer-Optimization Technique (POT)	Domain and Desired Objective	Model	Findings	Refs.
Prosumer Genetic Algorithm (PGA)	Domain: Residential energy management system. It is concerned with the cost of electricity and reducing the peak-to-average ratio.	A hybrid-renewable generation and ESS load were controlled and handled using a Genetic Algorithm (GA).	We are dividing devices into clusters. User comfort is not regarded.	[128,129]
Prosumer Mixed Integer Linear Programming (PMILP)	Domain: Energy-management System and Grid connection aimed to reduce the peak-to-average ratio cost using renewable energy.	The energy management approach for residential based on renewable energy.	Peak-to-average ratio cost reduction was achieved.	[130,131]
Prosumer Particle Swarm Optimization (PPSO)	Domain: Appliance scheduling. Aimed to reduce energy costs and improve consumer satisfaction.	Produce electricity and handle energy demand.	A reasonable deal was made regarding benefits and user comfort.	[132,133,134,135]
Prosumer Linear programming (PLP)	Domain: Residential energy management system. Tends to reduce the electricity bills and peak-to-average ratio.	Energy use scheduling to avoid maximum load times based on ESS.	Tasks are to use ESS off-grid and then modify them during peak hours.	[136,137,138,139,140,141]
Prosumer Integer Linear Programming (PILP)	Domain: Residential energy management system. Appliance scheduling.	A scheduling approach to controlling home and neighborhoods energy demand.	The integration and management of electricity usage trends are achieving a significant reduction in peak periods.	[142,143,144]

**Table 5 sensors-22-08397-t005:** Reported research on SG 2.0.

Refs.	Prosumer in SG 2.0			SG 2.0 Challenges		Blockchain/SG 2.0 Application Sector	
	SG 1.0	Micro-Gid	P2P	Real-Time	Virtual	Commercial	Residential
[22]		**τ**	**τ**	**τ**		**τ**	
[40]	**τ**	**τ**	**τ**				**τ**
[50,111]	**τ**	**τ**	**τ**				**τ**
[82]		**τ**					**τ**
[15,140]		**τ**			**τ**	**τ**	**τ**
[121]		**τ**	**τ**		**τ**	**τ**	
[31]	**τ**		**τ**			**τ**	**τ**
[25]	**τ**		**τ**		**τ**		**τ**
[28,103]			**τ**				**τ**
[19,64]		**τ**		**τ**		**τ**	**τ**
[37,100]		**τ**		**τ**	**τ**		
[29,115]		**τ**		**τ**	**τ**	**τ**	
[41,42,43,44,45,46,47,48,49,50,51,52,53,54,55,56,57,58,59,60,61,62,63,64,65,66,67,68,69,70,71,72,73,74,75,76,77,78,79,80,81,82,83,84,85,86,87,88,89,90,91,92,93,94,95,96,97,98,99,100,101,102,103,104]		**τ**		**τ**		**τ**	
[12,42]		**τ**		**τ**		**τ**	
[18,40,41,42,43,44,45,46,47,48,49,50,51,52,53,54,55,56,57,58,59,60,61,62,63,64,65,66,67,68,69,70,71,72,73,74,75,76,77,78,79,80,81,82,83,84,85,86,87,88,89,90,91,92,93,94,95,96,97,98,99,100,101,102,103,104,105,106,107,108,109,110,111,112,113,114,115,116,117,118,119,120,121,122,123,124,125,126,127,128,129,130,131,132,133,134,135,136,137,138,139,140,141,142,143,144]							
[16,119]							
[40,120]						**τ**	
[66,86]	**τ**					**τ**	
[107,135]	**τ**						
[16,87,91]							
[50,71]			**τ**				
[85]			**τ**				
[67,68,69,70,71,72,73,74,75,76,77,78,79,80,81,82,83,84]	**τ**	**τ**	**τ**	**τ**			
[102]		**τ**	**τ**	**τ**			**τ**
[110]	**τ**	**τ**		**τ**			**τ**
[49]	**τ**		**τ**	**τ**	**τ**		**τ**
[106]			**τ**	**τ**	**τ**		**τ**

**τ**: This topic has been cited in this/these reference(s).

## Data Availability

Not applicable.

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
