# Peer review of "Energy Internet Opportunities in Distributed Peer-to-Peer Energy Trading Reveal by Blockchain for Future Smart Grid 2.0"

_sensors, 2022, doi:10.3390/s22218397_

Round 1

Reviewer 1 Report

In this work, an overview of peer-to-peer energy exchange and how blockchain can be used to increase transparency and overall performance, including the degree of decentralization, scalability, and device reliability. After reviewing carefully, the reviewer found that the paper is well written. However, there are still some scopes to improve the paper further. The comments are as follows:

1. Avoid use of we/our/us.

2. The paper required proofreading for improving the technical quality of writing.

3. Table 2 should be shifted to page 10 from page 9. Because all the data of Table 2 is presented in page 10. Further in table 2, insert a separate column for references instead of writing the references in column 4 as like table 4.

4.      Write a section regarding block chains practical application in smart grid.

5.      Extend the future direction section further with appropriate example.

6.   The findings of the research paper are missing in conclusion. Please add findings.

Author Response

Original Article Title:Energy Internet Opportunities in Distributed Peer-to-Peer Energy Trading Reveal by Blockchain for Future Smart Grid 2.0: A Survey”

Re: Response to reviewers

Dear Editor,

Thank you for allowing a resubmission of our manuscript, with an opportunity to address the reviewers’ comments.

We are uploading (a) our point-by-point response to the comments (below) (response to reviewers), (b) an updated manuscript with yellow highlighting indicating changes, and (c) a clean updated manuscript without highlights.

Dear, we have revised our manuscript according to the concerns stated by the reviewers as explained below:

N.B: Grammar is cheeked and improved 

The paper title has been updated by: Energy Internet Opportunities in Distributed Peer-to-Peer Energy Trading Reveal by Blockchain for Future Smart Grid 2.0: A Survey

Reviewer #1: In this work, an overview of peer-to-peer energy exchange and how blockchain can be used to increase transparency and overall performance, including the degree of decentralization, scalability, and device reliability. After reviewing carefully, the reviewer found that the paper is well written. However, there are still some scopes to improve the paper further. The comments are as follows:

  1. Avoid use of we/our/us.
  2. The paper required proofreading for improving the technical quality of writing.
  3. Table 2 should be shifted to page 10 from page 9. Because all the data of Table 2 is presented in page 10. Further in table 2, insert a separate column for references instead of writing the references in column 4 as like table 4.
  4. Write a section regarding block chains practical application in smart grid.
  5. Extend the future direction section further with appropriate example.
  6. The findings of the research paper are missing in conclusion. Please add findings.

Reviewer #1:  Concern # 1:  1. Avoid use of we/our/us.

Author response:   all Sections have been updated and improved.

Author action: We have removed we/our/us from all Sections to update and improve them.

Reviewer #1:  Concern # 2:  The paper required proofreading for improving the technical quality of writing.

   Author response:   The paper is updated and improved.

Author action: All sections of the paper have been rephrased as we proofread it, and the grammar has been examined.

Reviewer #1:  Concern # 3:  Table 2 should be shifted to page 10 from page 9. Because all the data of Table 2 is presented in page 10. Further in table 2, insert a separate column for references instead of writing the references in column 4 as like table 4.

Author response:   Table 2 is updated.

Author action: Table 2 moved up to the asked page and a paragraph. In response to the review comment, we added a reference column.

Reviewer #1:  Concern # 4:  Write a section regarding block chains practical application in smart grid

Author response:   we added a section describing blockchain application

Author action: In response to the review comment, the paper is updated by adding a section 4: “Blockchain Application in SG 2.0”.

Reviewer #1:  Concern # 5:  Extend the future direction section further with appropriate example

Author response:   the paper is updated  

Author action:  Updates were made to the introduction, related literature, and conclusion. There are now two additional subsections, and they are : “ 5.1  SG 2.0 Values”, “SG2.0 Operational Challenges”.

  1. Reviewer #1:  Concern # 6:  The findings of the research paper are missing in conclusion. Please add findings.

      Author response:   The conclusion is updated  

Author action: The conclusion has been amended to include findings, future work, and limitations in response to the review comment.

Reviewer 2 Report

The article treats the interesting and topical matter of using blockchain for peer-to-peer energy trading in microgrids.

The article is structured correctly and the content is presented in a logically consistent order. However, there is a lack of the "Discussion and limitations" section.

There are two main drawbacks of the manuscript.

Firstly, the authors have not described the procedure of the review. I recommend referring to the paper: B. Kitchenham and S. Charters, ‘‘Guidelines for performing systematic literature reviews in software engineering,’’ Dept. Comput. Sci., Univ. Durham, Durham, U.K., EBSE Tech. Rep. EBSE-2007-01, 2007.

Secondly, the set of references lacks many important and up-to-date papers published in the MDPI. However, it should be emphasized that the authors have included one important paper by Yahaya et al. [96].

I urge the authors to include in the review the following papers that deal with prosumer communities, peer-to-peer trading, and smart contracts in energy sector applications:

- A Smart Contract-Based P2P Energy Trading System with Dynamic Pricing on Ethereum Blockchain, https://doi.org/10.3390/s21061985,

- Reconfigurable Smart Contracts for Renewable Energy Exchange with Re-Use of Verification Rules, https://doi.org/10.3390/app12115339,

- Blockchain Based Transaction System with Fungible and Non-Fungible Tokens for a Community-Based Energy Infrastructure, https://doi.org/10.3390/s21113822,

- Blockchain Technology Applied to Energy Demand Response Service Tracking and Data Sharing, https://doi.org/10.3390/en14071881,

- A New Vision on the Prosumers Energy Surplus Trading Considering Smart Peer-to-Peer Contracts, https://doi.org/10.3390/math8020235.

In the "References" section the authors should remove the bibliographic item [118]. It does not contain reference details (empty point). 

Another major disadvantage of the manuscript is an incomplete set of references. In the text of the manuscript, the authors cite papers up to number 155 (line 260). However, in the "References" section there are only 138 bibliographic items.

Moreover, some papers seem to be not relevant for the study, e.g. [129], [130]. I would like the authors to check again the relevance of references from 129 to 138.

The manuscript requires thorough refinement and I encourage the authors to do additional work.

In general, the manuscript looks as if it is half finished.

Author Response

Response to reviewers

Original Article Title:Energy Internet Opportunities in Distributed Peer-to-Peer Energy Trading Reveal by Blockchain for Future Smart Grid 2.0: A Survey”

Re: Response to reviewers

Dear Editor,

Thank you for allowing a resubmission of our manuscript, with an opportunity to address the reviewers’ comments.

We are uploading (a) our point-by-point response to the comments (below) (response to reviewers), (b) an updated manuscript with yellow highlighting indicating changes, and (c) a clean updated manuscript without highlights.

Dear, we have revised our manuscript according to the concerns stated by the reviewers as explained below:

N.B: Grammar is cheeked and improved 

The paper title has been updated by: Energy Internet Opportunities in Distributed Peer-to-Peer Energy Trading Reveal by Blockchain for Future Smart Grid 2.0: A Survey

Reviewer #2:  Concern # 2:  Secondly, the set of references lacks many important and up-to-date papers published in the MDPI. However, it should be emphasized that the authors have included one important paper by Yahaya et al. [96]. I urge the authors to include in the review the following papers that deal with prosumer communities, peer-to-peer trading, and smart contracts in energy sector applications:

  • A Smart Contract-Based P2P Energy Trading System with Dynamic Pricing on Ethereum Blockchain, https://doi.org/10.3390/s21061985,
  • Reconfigurable Smart Contracts for Renewable Energy Exchange with Re-Use of Verification Rules, https://doi.org/10.3390/app12115339,
  • Blockchain Based Transaction System with Fungible and Non-Fungible Tokens for a Community-Based Energy Infrastructure, https://doi.org/10.3390/s21113822,
  • Blockchain Technology Applied to Energy Demand Response Service Tracking and Data Sharing, https://doi.org/10.3390/en14071881,
  • A New Vision on the Prosumers Energy Surplus Trading Considering Smart Peer-to-Peer Contracts, https://doi.org/10.3390/math8020235.

Author response:   The paper has been improved and updated in accordance with the suggested references.

Author action:   The subsection :3.4. Smart Contracts in Energy Sector Applications” has been included to improve and update the document.

Additional references range from [99] to [106].

Reviewer #2:  Concern # 3:  In the "References" section the authors should remove the bibliographic item [118]. It does not contain reference details (empty point).

Author response:   The paper has updated in accordance with the suggested references.

Author action:   In accordance with the section’s objectives, the subsection has been revised: 4.2 “The Evolution and Structure of the Blockchain,” and reference [118] is modified.

Reviewer #2:  Concern # 4:  Another major disadvantage of the manuscript is an incomplete set of references. In the text of the manuscript, the authors cite papers up to number 155 (line 260). However, in the "References" section there are only 138 bibliographic items

Author response:   In compliance with the proposed suggestion, the paper has been revised.

Author action:   Updates have been made to the references table and paper section.

Reviewer #2:  Concern # 5:  Moreover, some papers seem to be not relevant for the study, e.g. [129], [130]. I would like the authors to check again the relevance of references from 129 to 138.

Author response:   All references have been updated and checked.

Author action:   In accordance with the section’s objectives, the subsection has been revised: Prosumer Mixed Integer Linear Programming

Reviewer #2:  Concern # 6:  The manuscript requires thorough refinement and I encourage the authors to do additional work. In general, the manuscript looks as if it is half finished

Author response:   the paper has been improved

Author action:   To improve the paper, some sections and table have been added. Each section has been improved and improved. The following sections have been added to further the objectives of the paper, such as:

  • Smart Grid 2.0 - SG 2.0 section
  • Smart contracts and application
  • SG 2.0 Values
  • 0 Operational Challenges
  • Smart Contracts in energy sector

Round 2

Reviewer 2 Report

The authors present a survey on applying blockchain for Future Smart Grid 2.0.

The article is structured correctly and the content is presented in a logically consistent order.

I confirm that the authors have addressed virtually all of my concerns. They completely rebuilt the "Introduction" section and significantly improved the "References" section according to all comments.

The manuscript has been improved precisely and the authors have clearly marked all changes. 

The English stylistic and punctuation errors have also been corrected thoroughly. 

I recommend accepting the paper in its present form.